# Microbial Dynamics during labneh Ambaris Production in Earthenware Jars

**DOI:** 10.3390/foods12163131

**Published:** 2023-08-21

**Authors:** Reine Abi Khalil, Christel Couderc, Sophie Yvon, Delphine Sicard, Frédéric Bigey, Gwenaelle Jard, Rabih El Rammouz, Pierre Abi Nakhoul, Hélène Eutamène, Marie-José Ayoub, Hélène Tormo

**Affiliations:** 1Department of Food Sciences and Technologies, Faculty of Agricultural and Veterinary Sciences, Lebanese University, Beirut 14-6573, Lebanon; reineabikhalil94@gmail.com (R.A.K.); pabinakhoul@ul.edu.lb (P.A.N.); 2Département des Sciences de l’Agroalimentaire et de la Nutrition, INP EI-Purpan, Université de Toulouse, 75 voie du TOEIC, BP 57611, F-31076 Toulouse, CEDEX 3, France; christel.couderc@purpan.fr (C.C.); sophie.yvon@purpan.fr (S.Y.); gwenaelle.jard@purpan.fr (G.J.); helene.eutamene@purpan.fr (H.E.); 3Toxalim, UMR 1331, INRA, INP-ENVT, INP EI-Purpan, Université de Toulouse, F-31027 Toulouse, France; 4SPO, INRAE, Institut Agro Montpellier, Université de Montpellier, F-34060 Montpellier, France; delphine.sicard@inrae.fr (D.S.); frederic.bigey@inrae.fr (F.B.); 5Department of Animal production, Faculty of Agricultural and Veterinary Sciences, Lebanese University, Beirut 14-6573, Lebanon; rabih.elrammouz@ul.edu.lb

**Keywords:** microbial dynamics, milk fermentation, DNA metabarcoding, goat’s milk

## Abstract

The responses of various microbial populations to modifications in the physicochemical properties of a food matrix, as well as interactions between these populations already present, are the main factors that shape microbial dynamics in that matrix. This work focused on the study of microbial dynamics during labneh Ambaris production, a traditional Lebanese concentrated fermented goat milk made in jars during 3 months. This was assessed in two earthenware jars at a production facility. DNA metabarcoding of the ITS2 region as well as the V3–V4 region of the 16S rRNA gene was used to characterize the fungal and bacterial communities, respectively. Viable bacterial isolates were also identified by Sanger sequencing of the V1–V4 region of the 16S rRNA gene. Our results showed that the dominant microorganisms identified within labneh Ambaris (*Lactobacillus kefiranofaciens*, *Lentilactobacillus kefiri*, *Lactococcus lactis*, *Geotrichum candidum*, *Pichia kudriavzevii* and *Starmerella* sp.) settle early in the product and remain until the end of maturation with varying abundances throughout fermentation. Microbial counts increased during early fermentation stage, and remained stable during mid-fermentation, then declined during maturation. While microbial compositions were globally comparable between the two jars during mid-fermentation and maturation stages, differences between the two jars were mainly detected during early fermentation stage (D0 until D10). No significant sensorial differences were observed between the final products made in the two jars. Neither coliforms nor *Enterobacteriaceae* were detected in their viable state, starting D7 in both jars, suggesting the antimicrobial properties of the product.

## 1. Introduction

Fermented milks result from the activities of microbial communities that could either naturally inoculate milk (spontaneous fermentation) or be intentionally added [1,2,3]. Dairy products obtained by spontaneous fermentations are usually colonized by indigenous microbial communities, which could have several origins such as the milk used, the production utensils, the surrounding dairy environment, and the human interventions. These fermentations often result in highly diversified and rich-microbial communities encompassing many bacterial and fungal species [4,5,6]. The ability of these microbial communities to colonize and survive in the food matrix is the result of their responses to various abiotic factors, such as the change in the physicochemical conditions of their habitat (pH, salt content, and water activity) and biotic factors such as their interactions with one another [7,8,9]. Traditional fermented milks are an essential part of the everyday diet of the Eastern Mediterranean populations [10]. They are primarily produced in rural areas, enhancing the social welfare and food security of those populations [11]. Current scientific knowledge of the main characteristics of the vast majority of these traditional fermented milks is rather limited, and even less is known about their microbial ecosystems, potential health effects, and sanitary aspects [10].

Labneh Ambaris is a traditional fermented raw goat’s milk product produced in Lebanon: a country that borders the Mediterranean Sea. It is widely made at household level in specific Lebanese rural regions, using a typical production process in special earthenware jars [12,13]. Briefly, the production process of labneh Ambaris lasts from 2 to 5 months and requires different steps characterized by regular whey drainages accompanied by additions of raw milk with coarse salt. When the jar is full, the product is given time to mature for 10 to 30 days. The content of the jar is then transferred to cloth tissues for additional draining, and the finished product is usually shaped into balls, stored in glass jars, or piled without or with olive oil for later consumption [12]. The earthenware jars used for the production of labneh Ambaris withstand productions for several seasons, but producers have to renew them regularly. For the communities involved in its production, both as consumers and sellers, labneh Ambaris is essential in enhancing food security. By offering a consistent source of nutrients, it not only serves the urgent nutritional needs of Lebanese rural inhabitants but also maintains the economic stability of these communities [12]. In previous studies, the microbial communities of labneh Ambaris samples collected from different producers at the end of maturation phase were identified [12,14]. These studies demonstrated using DNA metabarcoding and shotgun metagenomics techniques that the keystone microbial species at this stage of production were *Lactobacillus kefiranofaciens*, *Lentilactobacillus* sp. and *Lactococcus lactis* for bacteria, and *Geotrichum candidum*, *Pichia kudriavzevii* and *Kluyveromyces marxianus* for fungi. Additionally, a study revealed that, by using a culture-dependent method, lactobacilli isolated from labneh Ambaris may have probiotic and antibacterial properties [15]. However, even though a recent study monitored a labneh Ambaris production, mainly focusing on cultivable yeasts [13], the presence and dynamics of labneh Ambaris’ microbial communities throughout fermentation were not fully explored. Furthermore, it is crucial to investigate the dynamics of labneh Ambaris’ microbial communities throughout its long manufacturing process, in order to better understand the fermentation process and the resulting properties of labneh Ambaris.

Therefore, the main objective of our study was to explore the physicochemical characteristics and microbial dynamics during labneh Ambaris production from the initial raw milk until the final product throughout the three processing stages: early fermentation, mid-fermentation and maturation. Two productions each, in an earthenware jar having the same capacity, were monitored in parallel following the same production process. Culture-dependent and independent microbiological methods, in addition to physicochemical analysis, were used to study the microbial and physicochemical dynamics throughout the production, and finally, a sensory discriminating test was used to assess if any differences could be detected between the two finished products.

## 2. Materials and Methods

### 2.1. Labneh Ambaris Production and Sampling

The production facility chosen for our study is located in the Chouf district in Lebanon. During the previous production season (2020), labneh Ambaris samples were collected at the end of maturation stage from this facility and characterized [12]. This producer was selected due to their representation of the traditional production process and their commitment to maintaining good hygienic conditions. Their labneh Ambaris products from the previous season were tested and found free from any pathogens [12]. Moreover, the microorganisms found in their previous labneh Ambaris samples represent the keystone species identified for this product [12].

Two labneh Ambaris productions were initiated simultaneously from June until October 2021 in separate earthenware jars having the same capacity (15 Liters) and shape. The first one had already been used for labneh Ambaris production for a minimum of 5 previous seasons (referred to as Jar A), while the second one was new (referred to as Jar B). These two types of jars are typically found at a production facility since they can sustain production for numerous seasons but need to be replaced on a regular basis by producers. Jar A had already been washed with soap and water at the end of the previous season (2020) and allowed to dry until the following season, whereas Jar B was cleaned with soap and water and left to dry 1 week before using it in this production. Throughout the production process, samples were collected from both jars at the same time. Each time, 100 g of labneh Ambaris was taken from each jar in accordance with the sampling plan shown in Appendix A. A ladle sanitized with 70% ethanol was used for the collection, and it was performed at the same level each time inside the jar. Details of the production process, as well as the collected samples, are presented in Appendix A. With each raw milk addition, a fixed quantity of 500 g of coarse salt was added to each of the jars. The production process of labneh Ambaris was divided into three stages according to the steps involved: (1) the early fermentation stage (from D0 until D10), where raw goat’s milk and coarse salt were initially placed in the jar at D0 and left to ferment without interventions until the first whey drainage that was conducted at D10, indicating the end of this stage; (2) the mid-fermentation stage (from D10 until D93) characterized by regular whey drainages and additions of raw milks with coarse salt; and (3) the maturation stage that starts after the last whey drainage (from D93 until D103) and ends when the product is prepared for long term conservation.

All fermented milks collected from both jars were subjected to microbiological enumeration, pH measurements and salt contents determination. The initial raw milk (RM-D0) was subjected to microbiological enumeration and pH measurement. All collected samples (fermented milks and raw milks) were analyzed by culture-independent microbial analysis. Culture-dependent analyses were performed for samples collected from both earthenware jars for the following fermentation days: D0, D2, D4, D10, D47, and D100. The choice was based on the pH values measured during each stage, which we assumed best reflects the fermentation activity in the system. Sensory analysis was conducted for the finished products at D103 for both jars. The detailed analytical approach is shown in Figure 1.

### 2.2. Physicochemical Analyses

The pH values were recorded at room temperature for all fermented milks and the initial raw milk RM-D0 used a penetration probe (PHC 108, HACH, Manchester, UK) and a calibrated pH meter (HQ11 HD, HACH, Manchester, UK). Salt contents (%) were analyzed for all fermented milks according to NF EN ISO 5943 [16]. Briefly, 5 g of each sample was placed in a beaker. Then, 30 mL of pre-heated, 55 °C distilled water were added, followed by 2 mL of 4 mol/L nitric acid solution (CARLO ERBA, Rouen, France). The mixture was titrated with a 0.1 mol/L silver nitrate solution (Reagecon, Shannon, Ireland) using the 800 Dosino potentiometer titrator (Metrohm, Herisau, Switzerland) while constantly stirring until reaching the final point. The pH and the salt contents were performed in triplicates for each sample.

### 2.3. Microbiological Analyses

#### 2.3.1. Microbial Enumeration

To enumerate microorganisms, the collected samples (100 g) were directly placed at 4 °C until reaching the laboratory and were then used for the preparation of appropriate 10-fold decimal dilutions in sterile buffered peptone water (Scharlau, Barcelona, Spain). The diluted suspensions were plated on five nutritive and selective media: PCA (Liofilchem, VIA SCOZIA, Italy) for total mesophilic aerobic flora, MRS—de Man, Rogosa, and Sharpe agar (HIMEDIA, Mumbai, India) for mesophilic lactobacilli, and YPD—Yeast Peptone Dextrose agar for Yeast and molds (composition, as detailed by Abi Khalil et al. [12], and were each supplemented with 8 mL/L penicillin-streptomycin antibiotic solution 100X, BioWest, Nuaillé, France), VRBG (HIMEDIA, Mumbai, India) for *Enterobacteriaceae* and VRBL (HIMEDIA, Mumbai, India) for coliforms. For PCA, MRS and YPD, 0.1 mL of the dilutions were surface plated, and for VRBG and VRBL, 1 mL of the dilutions were mass plated. PCA and MRS media were incubated at 30 °C for 72 h, YPD at 25 °C for 5 days, and VRBG and VRBL at 37 °C for 24 h. For each sample, 2 series of dilutions were performed, each one plated on a Petri dish for each media. Results were expressed as log cfu/g of fermented milk samples and log cfu/mL of raw milk samples. After plating, sterile glycerol (20% [vol/vol]) was added to one of the two initial 10-fold dilutions and kept at −20 °C for bacterial species identification using a culture dependent technique described hereafter.

#### 2.3.2. Culture-Dependent Bacterial Species Diversity

##### Enumeration, Isolation and Identification of Presumptive Lactobacilli

The viable presumptive lactobacilli in the fermented milks were identified throughout fermentation in both earthenware jars based on the method described by Zanirati et al. [17] with modifications. Briefly, 0.1 mL of the serial dilutions were surface plated on MRS agar (Biokar, Beauvais, France) dissolved in lactic whey (lwMRS, pH 5.5) supplemented with natamycine (18 mg/mL of Delvo^®^Cid Instant, DSM, Maastricht, The Netherlands). The plates were then incubated at 30 °C for 14 days in a microaerophilic controlled atmosphere (BD GasPak EZ Campy Gas generating Pouch System, BD; Beton, Dickinson and Company, NJ, USA) containing between 5 and 15% oxygen. After incubation, the resulting colonies were enumerated and 10% of the total colony number of the plates with 30 to 300 colonies were isolated according to different morphologies. The isolated colonies were then cultivated in MRS broth (Biokar, Beauvais, France) at 30 °C for 48 h. The bacterial isolates were then purified 3 times on MRS agar (Biokar, Beauvais, France) supplemented with natamycine (18 mg/mL of Delvo^®^Cid Instant, DSM, Maastricht, The Netherlands), and were then subjected to Gram and catalase tests to identify the presumptive lactic acid bacteria (LAB) as Gram-positive or catalase-negative. All of the isolated purified colonies were stored at −20 °C in 80% MRS broth (Biokar, Beauvais, France) supplemented with 20% sterile mixture of glycerol (15%) and skim milk (85%).

##### LAB Identification Using 16S rRNA Gene Sequencing

The chosen purified isolates (146 isolates), representing all different morphologies, were cultured by inoculation: 50 µL of conserved tube in 5 mL of MRS broth (Biokar, Beauvais, France) and incubated at 30 °C for 48 h. Then, 1 mL of the overnight cultures were placed in a sterile tube and centrifuged to remove the supernatant. The bacterial pellet was then suspended in 200 µL sterile deionized PCR-grade water and vortexed to mix. The resulting suspension was used as the DNA for subsequent colony PCR. For the isolates where negative colony PCRs were obtained, the pellets were subjected to DNA extractions using DNeasy Blood and Tissue Kit (QIAGEN, Hilden, Germany). Presumptive LABs were then identified by sequencing of the V1-V4 region of the 16S rRNA gene. For each PCR tube representing one bacterial strain, the mix was made as follows: 1 µL DNA, 15.75 µL deionized water, 5 µL Q5 buffer reaction 5X (New England Biolabs, Ipswich, England), 0.5 µL dNTPS 10 mM, 1.25 µL Forward primer 10 µM (5′-GAGTTTGATCCTGGCTCAG-3′), 1.25 µL reverse primer 10 µM (5′-TGGACTACCAGGGTATCTAAT-3′) and 0.25 µL High-fidelity Taq polymerase (New England Biolabs, Ipswich, UK). PCR was performed with the following amplification program: 98 °C for 5 min, 30 cycles at 98 °C for 10 s, 63 °C for 30 s and 72 °C for 30 s, and 72 °C for 2 min. The resulting PCR products of approximately 806 bp were then purified before sequencing, according to the method of Sanger et al. [18]. The sequences were then determined using the BLASTN program with the standard databases (http://www.ncbi.nlm.nih.gov/BLAST, accessed on 14 October 2022).

#### 2.3.3. Culture-Independent Analyses: DNA Extraction, Metabarcoding Sequencing and Bioinformatics

The DNA metabarcoding analyses were performed as detailed in our previous study [12]. Briefly, the total DNA from each sample was extracted using the DNeasy PowerFood Microbial Kit (Qiagen, Hilden, Germany). DNA concentrations were then adjusted to 10 ng/µL. For bacterial and fungal sequencing, mock communities (a combination of equimolar DNA concentrations 10 ng/L of various identified species) were added, as previously described by Abi Khalil et al. [12], and supplemented with the species *Geotrichum candidum* in the fungal mock.

The bacterial and fungal diversities were evaluated by DNA metabarcoding sequencing of the amplified V3–V4 region of the 16S rRNA gene for bacteria and ITS2 region for fungi. These regions were amplified and sequenced by Illumina MiSeq, as described in our previous study [12]. All amplicons were purified, quantified and then pooled together to constitute the library, which was then sequenced with an Illumina MiSeq protocol generating 300 bp paired-end reads. The resulting sequences were analyzed consecutively using DADA2 [19] and FROGS [20] pipelines. The pre-processing steps were carried out using DADA2 package version 1.14.1 in R program version 3.6.1. This resulted in Amplicon Sequence Variants (ASVs) tables. Taxonomic affiliations using DAIRYdb database version 2.0 [21] for the 16S rRNA gene and UNITE database version 8.2 [22] for ITS2 sequences were conducted using FROGS version 3.2.2 [20]. Multi-affiliations were then checked using the affiliation Explorer [23]. Bacterial and fungal ASVs tables were rarefied while creating the Phyloseq Objects with FROGSSTAT Phyloseq Import Data (Galaxy Version 3.2.2) before assessing the samples’ diversities. Species diversity within each sample was estimated by the α-diversity indices “richness”, and Shannon using FROGGSTAT Phyloseq Alpha Diversity tool (Galaxy Version 3.2.2). Bray–Curtis dissimilarity indices were used to estimate species diversity between the samples. These indices were calculated using FROGGSTAT Phyloseq Beta Diversity tool (Galaxy Version 3.2.2). Hierarchical clustering of the fermented milks of both jars, based on the Ward linkage method and Bray–Curtis matrix, was then applied and visualized using the FROGSSTAT Phyloseq Sample Clustering tool (Galaxy Version 3.2.2).

### 2.4. Sensory Triangular Testing

A triangular test was performed to determine whether there are sensorial differences between labneh Ambaris produced in the two studied jars. The sensory analysis was approved by the Lebanese University and INP-EI Purpan Institutions and took place during the autumn season of 2021 in a dedicated room close to the panelists’ residences or work places. Twenty-five panelists, 15 women and 10 men between the ages of 24 and 65, were chosen. Panelists were informed about the products’ nature and provided written consent to take part in the analysis.

Before serving the panelists, the refrigerated samples were brought to room temperature. Finished labneh Ambaris samples (A-D103 and B-D103) were coded with 3-digit random codes and approximately five g of each sample were presented to the panelists as per randomized combination order. During the sensory session, each panelist was given three samples simultaneously (one different and two representing the same sample) and was given instructions to taste them in order from left to right then choose the different sample. They were asked to cleanse their palates with a small piece of Lebanese bread (pita) then with room temperature water.

Statistical tests. According to the ISO 4120:2021 [24] norm-sensory analysis methodology-triangle test, a minimum of 17 correct answers were required for a panel of 25 judges to say that the two samples are not significantly different, with a confidence interval of 99.99%.

## 3. Results

### 3.1. Evolution of the pH Values and Salt Contents during Labneh Ambaris Production

During the early fermentation stage (D0 until D10), pH measurements in both jars showed a rapid decrease only after 2 days of fermentation, dropping from 6.51 units at D0 to 3.55 units in Jar A and 3.79 units in Jar B (Figure 2A). Throughout the early fermentation stage, values kept dropping until they reached their lowest points at the start of the mid-fermentation stage at D29 (3.14 for the Jar A and 3.19 for the Jar B, respectively), after which they remained almost stable until the end of maturation stage.

Salt contents (Figure 2B) were higher in Jar B than in Jar A. The values in both jars increased gradually until reaching their maximum at D47 for Jar A with 7% and at D93 for Jar B with 7.75%. The values then decreased until reaching at the end of maturation stage at D103, 2.75% and 3.79% for Jar A and Jar B, respectively.

### 3.2. Microbial Dynamics during labneh Ambaris Production

#### 3.2.1. Over Time Variation of Microbial Counts

Microbial population counts were monitored on five nutritive and selective media from D0 until D103 during labneh Ambaris productions, made in two earthenware jars. The microbial counts obtained are detailed in Appendix A. The initial raw milk RM-D0, used to start both productions, harbored high-total mesophilic aerobic flora as well as yeast and fungal populations (6.26 log cfu/mL and 5.10 log cfu/mL, respectively). Lower counts of presumptive lactobacilli were detected with 4.46 log cfu/mL. Bacterial species belonging to the *Enterobacteriaceae* family, as well as coliforms, were detected in comparable concentrations with 4.72 log cfu/mL and 4.82 log cfu/mL, respectively, in RM-D0. Throughout the fermentation process, the groups of microorganisms presented in Figure 3 mainly followed the same trend in both jars. For the three microbial groups’ total mesophilic aerobic flora (Figure 3A), presumptive lactobacilli (Figure 3B) and yeasts and molds (Figure 3C) during the early fermentation stage, microbial counts increased in both jars with noticeable fluctuations. The values remained globally stable during the mid-fermentation stage and showed a decrease during the maturation stage. Concerning the coliforms and the species belonging to the *Enterobacteriaceae* family, counts were two logs higher in Jar A at D2 compared to Jar B (Appendix A). Coliforms were not detected in their viable state, starting at D4 in the Jar A samples and D7 in Jar B, whereas species belonging to the *Enterobacteriaceae* family were not detected in both jars starting at D7.

#### 3.2.2. Microbial Diversity Dynamics throughout Fermentation Stages: α and β Indices

After quality filtering, denoising and merging, the V3–V4 region of the 16S rRNA gene and the ITS2 sequencing analyses resulted in a total of 1,209,227 reads with an average of 50,384 reads per sample for 16S, and 514,220 reads with an average of 21,425 reads per sample for ITS2. Final sequences (811,644 reads for 16S and 509,662 reads for ITS2) were then clustered into 509 and 50 Amplicon Sequences Variants (ASVs) for 16S and ITS2, respectively, after chimera removal. After normalization by rarefaction, 20,204 and 9663 sequences were kept per sample for 16S and ITS2, respectively. Alpha diversity indices were calculated based on ASVs tables for the fermented milks collected from both earthenware jars from D2 to D103, as well as the initial raw milk RM-D0 to investigate the bacterial and fungal diversity dynamics during production (Figure 4).

Concerning bacterial metrics, the initial raw milk (RM-D0) showed the highest richness index with 228 ASVs (Figure 4A) representing 103 species. The richness indices greatly dropped within 2 days of fermentation in both jars and remained stable until the maturation stage (Figure 4A). The indices of the samples collected from both jars showed mainly similar trends throughout the production process with slightly lower values in Jar A than Jar B. Indeed, by the end of mid-fermentation at D47, richness indices were almost double in Jar B (62 ASVs representing 42 species) than Jar A (31 ASVs representing 21 species). The differences remained almost the same between both jars until the end of the maturation stage, where 18 species and 37 species were detected in A-D103 and B-D103, respectively. The highest Shannon indices were recorded for the initial raw milk RM-D0 with a value of 2.71. Same for the richness indices, Shannon indices showed an overall decrease during fermentation in both jars. Overall, lower Shannon indices were observed in the Jar A samples throughout all fermentation stages with their minimum in mid-fermentation D29 and D47 (0.42 and 0.36, respectively vs. 2.30 and 1.63 for B-D29 and B-D47, respectively), meaning that, during this stage, the samples were mostly dominated by fewer bacterial species. Higher Shannon indices were recorded for the Jar B samples, meaning that higher bacterial diversity was observed in the Jar B samples compared to Jar A, mostly during the early fermentation stage.

As for the fungal metrics, they followed opposite trends compared to bacterial ones. The richness indices greatly increased following D2 of fermentation (from 2 in RM-D0 to 17 and 20 in A-D2 and B-D2, respectively). No particular changes were noted between the stages of fermentation within the same jar and the indices showed the same trends in both jars (Figure 4C). Slightly higher indices for the samples collected from Jar B were recorded, except for D93, where the richness index for A-D93 was 19 vs. 10 for B-D93. As for Shannon indices, the values increased during fermentation in both jars. The minimum values were recorded for samples B-D4 and A-D10 with 0.72 and 0.68, respectively; each of them were mainly dominated by one fungal species. The maximum values were recorded for samples B-D98 and A-D100 with 1.44 and 1.75, respectively.

Based on Bray–Curtis dissimilarity indices for bacterial communities, the Hierarchical Ascendant Classification (HAC) revealed that fermented milks were mainly clustered according to the fermentation stages (Figure 5). The bacterial clustering showed that, during the early fermentation stage, the Jar B samples clustered together and were remote from the Jar A samples that constituted a group of their own, meaning that the bacterial abundances were different at this stage between the two jars. As fermentation continued, samples of both jars were clustered in the same group for the mid-fermentation stage, except for sample B-D29. For the maturation stage, samples were clustered together independently of the jar, showing that the bacterial abundances were similar for the two jars, especially for samples collected at D98, D100 and D103 (Figure 5A). For fungal communities, clustering according to fermentation stages was less obvious than that of the bacterial communities (Figure 5B). However, samples collected from Jar A during the maturation stage clustered together. Additionally, most samples collected from both jars during the early stages of fermentation were pooled together (B-D2, A-D2, A-D7, A-D10 and B-D4). Samples taken in the middle of the fermentation process were dispersed among the other samples.

#### 3.2.3. Microbial Dynamics and Species Identification throughout Fermentation

Bacterial compositions analyzed by DNA metabarcoding showed that raw milk samples were dominated by the bacterial genera *Enterobacter* (22.84% of total sequences), *Chryseobacterium* (16.00%), *Acinetobacter* (11.46%), *Mycoplasma* (9.44%) and *Enterococcus* (8.72%). Concerning all the fermented milk samples, the most abundant bacterial genera were *Lactobacillus* (62.73% of total sequences), followed by *Lentilactobacillus* (17.87%), *Lactococcus* (7.78%), *Citrobacter* (2.17%) and *Enterobacter* (2.02%). Some species of *Lentilactobacillus* genus could not be identified and provided several options when blasted into the actual databases: *kefiri*/*otakiensis*/*sunkii* or *hilgardii*/*diolivorans*. The three bacterial species, *Lactobacillus kefiranofaciens*, *Lentilactobacillus* sp. and *Lactococcus lactis*, were present in almost all the samples but with varying abundances. It was noticeable that the bacterial composition of raw milks differed from that of the fermented milks (Figure 6). The 10 most abundant bacterial species within the raw milks and fermented milks of each jar are illustrated in Figure 6.

As for the culture-dependent method, a total of 146 bacterial isolates were collected from the raw and fermented milks from both jars and were identified based on the current databases. *Lentil. kefiri* was the dominant species in both jars throughout the production process. It was either accompanied by *Lb. kefiranofaciens* or *Lentilactobacillus diolivorans*, or both at the same time. The identified bacterial isolates selected from the initial raw milk RM-D0 and at some points throughout fermentation (D0, D2, D4, D7, D10, D47 and D100) are presented in Appendix A.

Concerning the fungal communities, regardless of the jar, the most abundant fungal genera were *Geotrichum* (55.54% of total sequences), followed by *Pichia* (20.60%), *Starmerella* (10.12%), and *Moniliella* (9.53%). Raw milk samples were dominated by *Geotrichum* with 77.56% of total sequences, followed by *Diutina* (9.07%) and *Starmerella* (3.58%) genera. It was noticeable that neither *Pichia kudriavzevii* nor *Moniliella* sp. were detected in any of the added raw milks while they were present in the fermented milks. The 10 most abundant fungal species within the raw and fermented milks of each jar are illustrated in Figure 7.

##### Early Fermentation Stage

During this stage (D0 until D10), bacterial compositions within each jar had similar compositions and relative abundances between the time points. When compared to one another, the samples collected from both jars showed differences mainly in the relative abundances of the detected species. The initial raw milk, RM-D0, was dominated by the bacterial species *Mycoplasma bovis*, *Chryseobacterium indoltheticum* and *Acinetobacter johnsonii* (presented in the “others” in Figure 6). *Lb. kefiranofaciens* and *Lentilactobacillus* sp. were detected in very low abundances in RM-D0 (0.19% and 0.05%, respectively), and *Lc. lactis* was not detected at all, but their relative abundances quickly increased within 2 days of fermentation in both jars. *Lb. kefiranofaciens* quickly became the dominant species within Jar A from D0 until D2 and remained as such until D10, whereas it was detected in much lower percentages in Jar B (72.89% in A-D2 vs. 29.69% in B-D2) during this fermentation stage. *Lentilactobacillus* sp. and *Lc. lactis* were present in both jars with comparable relative abundances. *Citrobacter freundii* was detected in higher abundances in the Jar B samples at D2 (1.65% relative abundance in A-D2 vs. 13.78% relative abundance in B-D2), and these abundances remained higher in Jar B throughout this fermentation stage. As for the identified isolates, the microbial counts were identical in both jars for all stages except A-D7, where no counts were detected (Appendix A). More than 50% of the isolates in RM-D0 were either identified as *Enterococcus faecium* or co-assigned to the *Enterococcus durans* species. The remaining 50% of the isolates were identified as *Lentil. kefiri*. This species was the main identified one during the early fermentation stage, noting that it was present at all time points and always representing a minimum of 50% of the identified isolates except for D10, where it was in comparable presence along with *Lb. kefiranofaciens* in both jars.

Concerning fungal communities analyzed by DNA metabarcoding, the initial raw milk RM-D0 was dominated by *Geotrichum candidum* with more than 96% of relative abundance and *Starmerella* sp. with 3.8% relative abundance. *G. candidum* was the dominant fungal species in all fermented milks during early fermentation stage with almost stable abundances between time points within the same jar. During the early fermentation stage, *Torulaspora delbrueckii* was detected in higher abundances in the Jar A samples (9.32% in A-D2) compared to Jar B (0.62% in B-D2), whereas *Trichosporon asahii* had higher relative abundances in the Jar B samples compared to Jar A (15.87% in B-D2 vs. 1.77% in A-D2). *Moniliella* sp. was detected in almost all fermented milks starting at D2. *Pichia kudriavzevii* was sporadically detected during this stage in samples A-D4, B-D7 and B-D10 (Figure 7).

##### Mid-Fermentation Stage

Throughout this stage (includes samples collected at D29 and D47), the bacterial compositions and relative abundances in the Jar B samples changed from one time point to the other, according to the addition of raw milks that were dominated by *Enterobacter asburiae* and other Gram-negative bacteria. Indeed, after each addition of raw milk, *Enterobacter asburiae*’s relative abundances increased in the Jar B samples compared to Jar A (11.03% in B-D29 vs. 0.13% in A-D29). Contrarily, the compositions of the Jar A samples were mostly stable during this stage, always having *Lb. kefiranofaciens* as the dominant species (more than 90% relative abundances in A-D29 and A-D47) accompanied by *Lentilactobacillus* sp. (Figure 6A). The species *Lc. lactis* was almost not detected in the Jar A samples during mid-fermentation stage, whereas it was present in the Jar B samples with 28% and 7.6% relative abundances at B-D29 and B-D47, respectively. As for the identified isolates from D47 samples, *Lentil. kefiri* was still the most dominant identified species in A-D47 with 84.62% of identified isolates, whereas *Lb. kefiranofaciens* was the abundant one in B-D47 with 47% of isolates (Appendix A).

As for fungal communities identified by DNA metabarcoding during this stage, *G. candidum* was identified in all samples with varying abundances and in the added raw milks with high abundances. *P. kudriavzevii* was detected in a very high abundance in A-D29 (almost 78% relative abundance), but not in the previous time point A-D10, noting that it was not detected in RM-D16. *Starmerella* sp. was detected for the first time in the mid-fermentation samples of both jars, whereas it was not detected in the early fermentation stage.

##### Maturation Stage

Overall, the bacterial compositions in both jar samples were comparable during this stage, except for the abundance of *Lc. lactis* that showed high values in the Jar B samples (especially B-D93 with 21.06%), while it was not detected in the Jar A samples. Apart from *Lc. lactis* in Jar B, the bacterial species *Lb. kefiranofaciens* and *Lentilactobacillus* sp. dominated the samples during this stage. On lwMRS culturing media, *Lentil. kefiri* was still the dominant species identified along with *Lb. kefiranofaciens* (in A-D100) and *Lb. kefiranofaciens* and *Lentil. diolivorans* (in B-D100).

Concerning fungal communities, the composition of samples collected from both jars were comparable and almost stable until the end of fermentation. Overall, the same fungal species were identified in both jars during the maturation stage except for *P. kudriavzevii*, which was not detected in the final product A-D103, but was present in B-D103 with 21.97% relative abundance, and *Pichia membranifaciens* species were detected for the first time in samples A-D100 and A-D103 (Figure 7).

### 3.3. Sensory Triangular Test: Discrimination between Jar A and Jar B

Out of the 25 judges, 11 provided correct answers (<17 correct answers) and were able to choose the different sample, whereas the remaining 14 could not. This means that the labneh Ambaris products made in Jar A and Jar B were not significantly different in terms of sensory features, with a 99.99% confidence interval.

## 4. Discussion

The traditional production of labneh Ambaris is a long process that takes place in earthenware jars, lasts for more than two months and requires regular additions of raw goat’s milk and coarse salt. In this study, we followed the physicochemical and microbial dynamics, occurring in two earthenware jars, throughout the production process of labneh Ambaris. The production processes were similar in both jars and the same milks as well as salt quantities were regularly added to the jars. Our results showed that, in both jars, the physicochemical parameters and microbial communities followed different trends according to the production stage (early, mid and maturation). A few differences were noticed between the jars’ microbial compositions and physicochemical parameters, mainly during the early fermentation stage. However, as the process went on, the ecosystems seemed to stabilize and the differences receded, which led to the end-products having comparable characteristics.

Our study confirmed that high acidity characterizes labneh Ambaris, whether during its production, or in the finished products, as shown in previous recent studies [12,13,25]. Very rapidly (within two days of fermentation), there was a sharp decline in pH in both jars that matched the rapid increase in microbial counts, especially presumptive lactobacilli and yeasts, and was consistent with previous observations made during the production of labneh Ambaris [13]. These low pH values could be attributed to various associations between microbial species, like that of *Lb. kefiranofaciens* and *Lc. lactis* [26], which dominated the samples early during fermentation. In addition, the association of LAB and yeasts, such as *P. kudriavzevii*, which was a dominant species within our samples, could also contribute to the low pH values obtained, as it was recently demonstrated [27]. The pH values remained stable during mid-fermentation until the end of maturation stage, which was concurrent with the stability observed in the microbial counts and is also consistent with previous observations [13].

Coarse salt that was placed at D0 in each jar was gradually dissolving into the fermented milks during early fermentation stage, which explains the increase in salt contents even though no salt additions were made during this stage. Salt contents in labneh Ambaris were shown to be highly dependent on the quantities added by the producer during the process [13]. This explains the varying but globally increasing salt contents detected during the mid-fermentation stage as regular whey drainages, and additions of milks with increasing coarse salt contents were made by the producer. As the maturation phase began on D93, and since no salt was further added during this stage while whey was drained until the end of the process, a gradual decrease in salt concentrations of fermented milks was observed. Due to the salt’s ability to dissolve in liquid media, some of it was expelled in the drained whey, while the remaining was kept in the trapped whey in curds.

A drastic decline in bacterial diversity was observed within two days of fermentation. Our results showed that the initial raw milk at D0 was mostly dominated by Gram-negative bacteria compared to lesser detected LAB. Raw milk can harbor a wide diversity of microorganisms from different origins [8,28] but only the species that can withstand the physicochemical characteristics of labneh Ambaris will be selected to remain and thrive within the product. No viable coliforms nor *Enterobacteriaceae* species were detected on the culturing media starting at D7 of fermentation for the two jars, despite the fact that the raw milks added to both jars, throughout the production, harbored a significant amount of *Enterobacteriaceae* species. The relatively prolonged contact time in high acidity and salt content conditions in labneh Ambaris could regulate these microorganisms’ growth and activity [4,29,30]. In addition, LAB are known to have inhibitory effects on bacteria, especially *Enterobacteriaceae* species by the secretion of bacteriocins [31] and other antibacterial compounds like peroxides, alcohols, and carbon dioxide [30,32]. Not detecting coliforms and *Enterobacteriaceae* species in our study contrasts to what was observed in another labneh Ambaris production, where numerous milk additions contributed to high *Enterobacteriaceae* contamination throughout fermentation [13]. It is worth noting that milk additions were not frequent during the process (only three were performed), which did not regularly feed *Enterobacteriaceae* or coliforms into the system. Consequently, and while Gram-negative bacteria, including *Enterobacteriaceae* and coliforms were the major group detected in raw milk at D0, their quick decline explains the drastic drop of bacterial diversity during the early stage. This decline might have masked any potential diversification that might have occurred in relation to other bacterial species, such as LAB. Bacterial diversity then remained stable during mid-fermentation and maturation stages in relation to the LAB species which stabilized and became established in the samples. To further investigate the pathogen-free nature of labneh Ambaris, conducting challenge tests by inoculating selected pathogenic or spoilage bacteria into labneh Ambaris samples would be worthwhile, as it would provide valuable insights into how the system responds and handles these potential contaminants. On the other hand, the diversity of yeasts and molds increased importantly, almost doubling, in both jars, within 2 days of fermentation in comparison to the initial raw milk (2 AVSs in RM-D0 to 17 ASVs and 20 ASVs in A-D2 and B-D2, respectively). Yeasts, which are not usually as numerous as bacteria in raw milk [33], might have originated from other sources to enrich the product. The low pH and high-salt conditions, well tolerated by yeasts [33,34,35,36], should have allowed them to thrive and be maintained during the production.

Among the dominant bacterial and fungal species that were rapidly detected in the fermented milks, only *Lentil. kefiri*, *Lb. kefiranofaciens*, *G. candidum* and *Starmerella* sp. were identified in the initial raw milk at very low percentages. This suggests that other sources than raw milk contribute to inoculating the fermented milks. Indeed, it was shown that microorganisms could be introduced into dairy products from an in-house microbiota already established within the production environment (on utensils for example) or from human interventions [5,6,36,37]. Furthermore, DNA metabarcoding revealed a much higher abundance of *Lb. kefiranofaciens* in Jar A than in Jar B as early as D2, while both jars had comparable viable counts of presumptive lactobacilli. It is worth mentioning that this was observed despite the fermentation system being closed, undisturbed and no milk additions made during the early fermentation stage. In our case, one of the in-house sources of inoculation might be the inner surface of the earthenware jars. Indeed, we can hypothesize that through the development of biofilms, microorganisms such as *Lb. kefiranofaciens* (which was more abundant since the start of fermentation in Jar A that was already used for previous labneh Ambaris productions) can remain from earlier productions on the inner surfaces of the containers, and when the conditions are ideal for growth, the microbial cells enclosed in their matrix can be released into the recently added milk [38,39,40]. Additionally, during mid-fermentation and maturation stages, *Moniliella* sp. was detected in high abundances in the fermented milks, while it never appeared in the raw milks. Interestingly, an observation reported in a previous study of ours might consolidate the hypothesis of an established in-house microbiota. Indeed, in our previous study, three labneh Ambaris samples were collected during a previous production season from the same producer’s facility where the current productions of this study were conducted [12]. The results of that study showed that, out of all collected samples from 17 different producers, a unique and distinctive presence of *Moniliella* sp. was recorded within all samples collected from this same producer [12].

After the early fermentation stage, the dominant bacterial (*Lb. kefiranofaciens*, *Lentil. kefiri* and *Lc. lactis*) and fungal (*G. candidum*, *P. kudriavzevii* and *Starmerella* sp.) species remained rather stable until the end of maturation stage, even if some variations in abundances were recorded. Once fermentation was initiated, it can be assumed that these dominant microorganisms, already installed in high numbers in the previous coagulum remaining in the jars after each whey drainage, act as natural starters that rapidly initiate the fermentation of the newly added milks and dominate the matrix. All precautions taken, this phenomenon of inoculation of new milks can recall the one that occurs in kefir, a popular fermented milk, where kefir grains are used as natural starters that initiate new fermentations and are then collected again at the end of fermentation to inoculate the following batch.

Even though both culture-dependent and independent methods successfully identified the two dominant bacterial species present at high abundances in both jars, some discrepancies were found between the two approaches. *Lentil. kefiri* was the most frequently found species on lwMRS media, whereas *Lb. kefiranofaciens* was the dominant species in almost all samples using DNA metabarcoding. Due to its demanding growth conditions, *Lb. kefiranofaciens* cannot be easily grown through conventional methods [41]. However, the two methods are also complementary. Indeed, the culture-dependent method may allow us to suppose that the co-assigned *Lentilactobacillus* sp. that could not be identified with DNA metabarcoding technique could be the species *kefiri* or/and *diolivorans*. Using the culture-dependent approach, a large collection of indigenous bacterial isolates was constituted. Therefore, characterizing these strains could help to better understand their metabolic potential, and maybe use them to constitute starter cultures for labneh Ambaris. To further examine the diversity of living bacteria and fungi, it would be interesting to identify isolates from other culturing media.

It is undeniable that dairy products’ sensory characteristics are strongly determined by the microbial populations they harbor. The fact that both productions of labneh Ambaris had very similar microbial compositions (especially after the early fermentation stage) could explain that no significant differences were detected between the two jars’’ finished labneh Ambaris samples. Even if Jar B was 1% higher in salt content, this was not a discriminating factor for the panelists. Salt-acid interactions occur in the human palate upon food consumption. Many studies have shown that acidity inhibits salty taste [42,43], knowing that both labneh Ambaris samples had a similar pH of 3.3 units. To complete this first sensorial result, it could be interesting to study the aromatic compounds in labneh Ambaris and their relationship with the microbial populations present in it.

Despite the differences recorded between the two fermentations, such as the salt contents or the use of a new jar, the microbial ecosystems were found to be similar at the end of production. To confirm or deny the impact of the jar’s age on the microbial profiles in this production facility, the experiment should be repeated and include at least three jars of each type. It would also be interesting to extend this work to other production facilities. Indeed, as shown in our previous study, the microbial profiles of labneh Ambaris at the end of maturation can vary from one facility to another [12]. Thus, to delve further into the study of microbial dynamics over time, especially their function, it would also be of interest to study the metabolites that are present in labneh Ambaris, such as alcohols, esters and aldehydes [44,45,46]. Then, the construction of simplified synthetic consortia can be proposed to better decipher the functioning of these communities. This approach aims to constitute a mix of several species (bacterial and fungal) in controlled conditions, allowing the study of their interactions and functions.

## 5. Conclusions

Overall, our results highlighted the microbial dynamics during labneh Ambaris production using both culture-dependent and independent microbiological methods. *Lb. kefiranofaciens*, *Lentil. kefiri*, *Lc. lactis*, *G. candidum*, *P. kudriavzevii* and *Starmerella* sp. were identified as the dominant bacterial and fungal species. They settled early in labneh Ambaris and remained until the end of the process with varying abundances according to the fermentation stage (early, mid and maturation). As fermentation continued, the systems seemed to stabilize and the final products had equivalent features. Labneh Ambaris was found to be free of spoilage and pathogenic microorganisms from the very first stage of production, despite the regular additions of raw milks that are typically dominated by these types of microorganisms.

## Figures and Tables

**Figure 1 foods-12-03131-f001:**
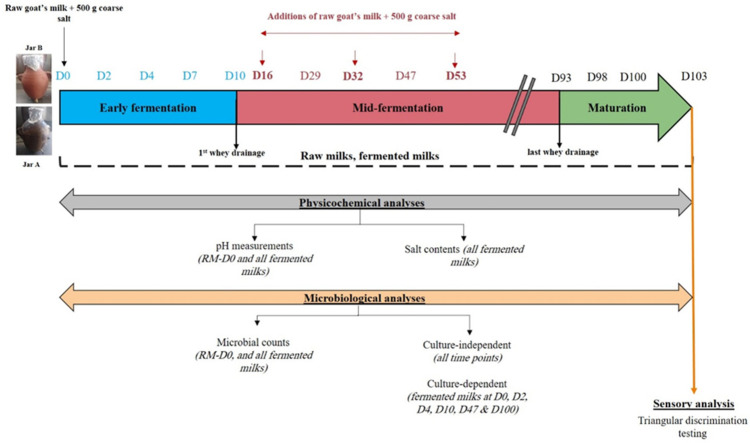
Analytical approach for the analyses of 2 labneh Ambaris productions during fermentation in earthenware jars A and B.

**Figure 2 foods-12-03131-f002:**
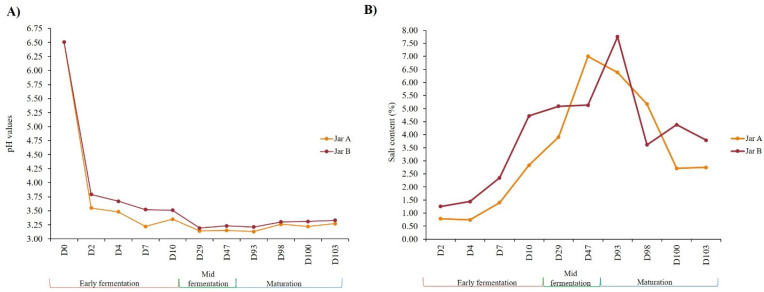
Evolution of the pH values (**A**) and salt contents (**B**) of labneh Ambaris samples collected throughout production from two earthenware jars A and B. The values showed here represent the average of the triplicates (n = 3) for pH and salt contents. Standard deviations were <1% for pH and <2% for salt contents and therefore were not shown on the figures.

**Figure 3 foods-12-03131-f003:**
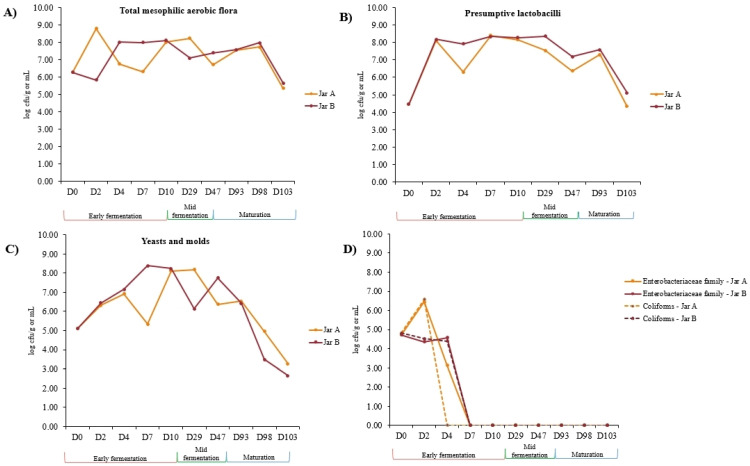
Microbial counts obtained on five nutritive and selective media for (**A**) Total mesophilic aerobic flora, (**B**) presumptive lactobacilli, (**C**) yeasts and molds, (**D**) *Enterobacteriaceae* family and coliforms, during production of labneh Ambaris from D0 until D103. The counts are expressed as the average (n = 2) log cfu/g of fermented milk samples and log cfu/mL of raw milk samples. All the values recorded are available in Appendix A.

**Figure 4 foods-12-03131-f004:**
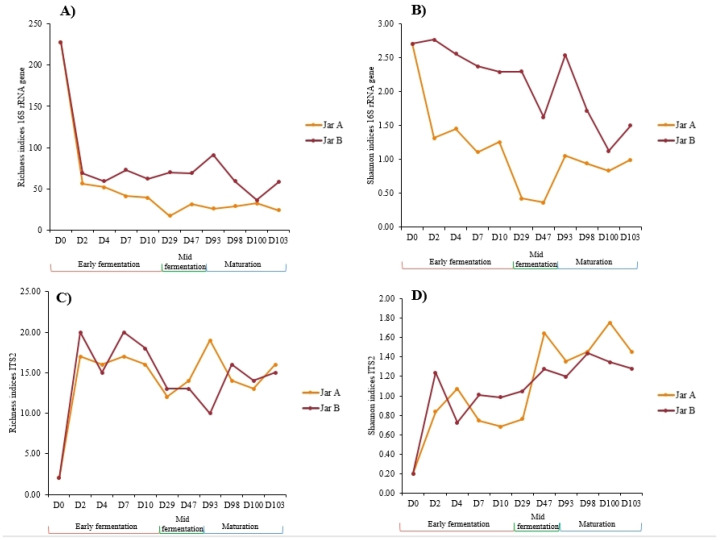
Bacterial (**A**,**B**) and fungal (**C**,**D**) metrics of alpha diversity indices for labneh Ambaris samples collected at different production stages from two earthenware jars: A and B.

**Figure 5 foods-12-03131-f005:**
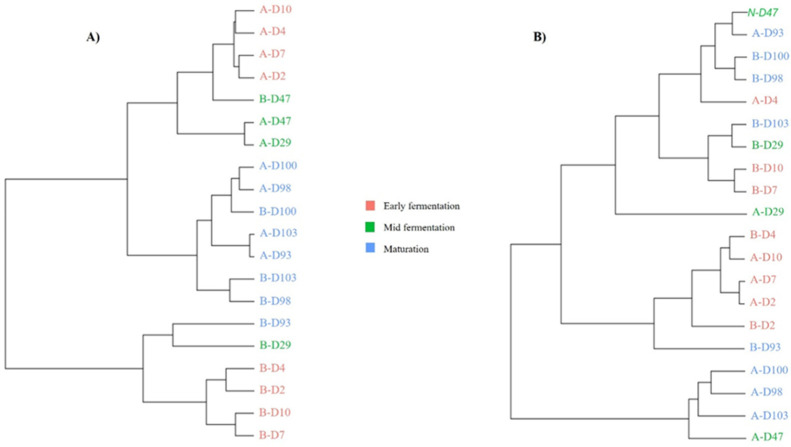
Hierarchical Ascendant Classification (HAC) based on Bray–Curtis dissimilarity indices for bacterial (**A**) and fungal (**B**) communities of labneh Ambaris samples collected from 2 earthenware jars A and B throughout production. Samples are color-coded according to the fermentation stages: pink for early fermentation (D2 to D10), green for mid-fermentation (D29 to D47) and blue for maturation (D93 to D103).

**Figure 6 foods-12-03131-f006:**
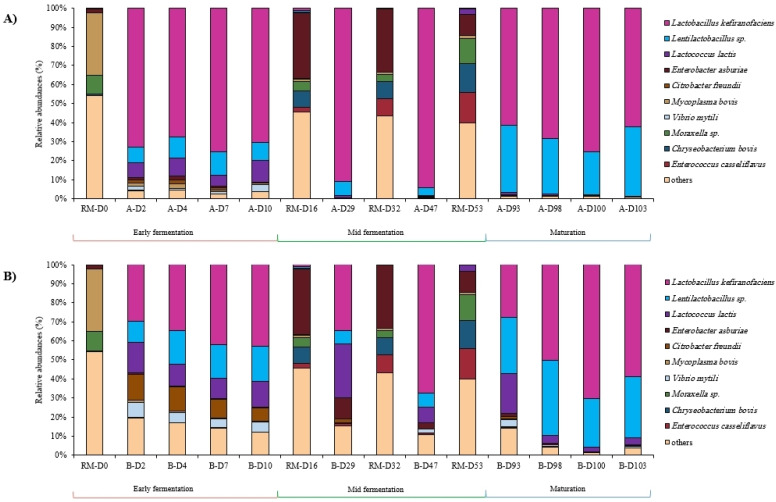
Bacterial communities’ dynamics in labneh Ambaris samples collected during production from 2 earthenware jars, jar A (**A**) and jar B (**B**), in addition to the raw milks added during production by sequencing of the V3–V4 region of the 16S rRNA gene.

**Figure 7 foods-12-03131-f007:**
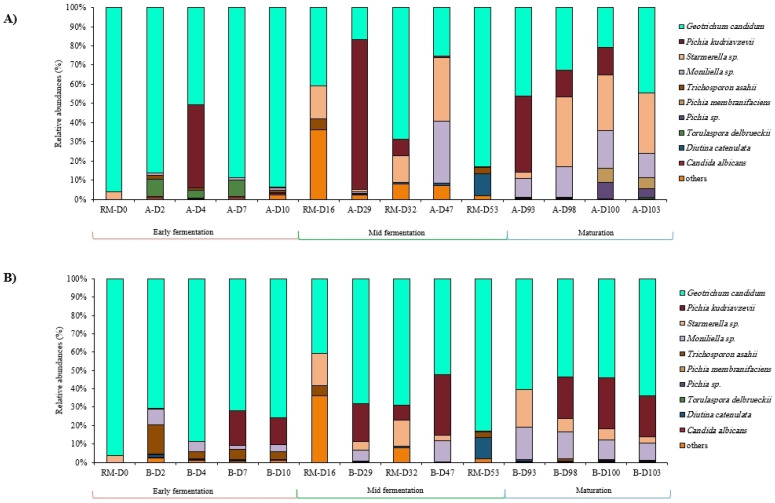
Fungal communities’ dynamics in labneh Ambaris samples collected during production from 2 earthenware jars, A (**A**) and B (**B**), in addition to the raw milks added during production by sequencing of the ITS2 region.

## Data Availability

The sequencing data generated in this study were deposited in the European Nucleotide Archive (ENA) under accession number PRJEB57561, and the other data used to support the findings of this study can be made available by the corresponding author upon request.

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
