# Peer review of "Microbial Dynamics during labneh Ambaris Production in Earthenware Jars"

_foods, 2023, doi:10.3390/foods12163131_

Round 1
Reviewer 1 Report
- some recent references for labneh is need
- some methods do not have sample preparation.
- any limitation of the study? please comment.
Reviewer 2 Report
Thanks authors for their research work. Traditional fermented milk could be a good source of health-beneficial bacteria. However, Some issues need to be addressed. Authors counted mesophilic bacteria, yeast and mold, also lactobacilli. I can see for some cases the yeast and mold count is higher than the mesophilic count (Fig 2). What could be the reasons? Also, the authors incubated the samples in microaerophilic conditions, did the authors count the anaerobic bacteria?
For sample B(Jar B), Can you please describe the sample preparation steps clearly?
So far I can understand Labneh Ambaris is a specific location-oriented fermented product, What is the significance of the product irrespective of the region?
The language of the manuscript is acceptable.
Reviewer 3 Report
In this manuscript, Khalil and co-workers studied the microbial dynamics during labneh Ambaris production in the jars for 3 months. They showed that the dominans microbes were Lactobacillus kefiranofaciens, Lentilactobacillus kefiri, Lactococcus lactis, Geotrichum candidum, Pichia kudriavzevii and Starmerella sp. and changes dynamically in the different phases of fermentation process. These can be interests for Readers. The manuscript still contains some errors required Authors to address. The introduction is quite general and lack of the gap that this work will fulfill. The names of microbes should be written in Italic form (Session 3.2).
See comments to Authors
Reviewer 4 Report
the study should have other indices (metabolites) to evaluate the product other than few chemical like pH & sensory parameters
Minor edit on language
Reviewer 5 Report
The authors presented interesting research but my main concern is why the research was not performed in triplicate. Can you give me the explanation how can we take these results as relevant when beeing performed for only 2 different yars.
Reviewer 6 Report
The paper “Microbial dynamics during labneh Ambaris production in earthenware jars” addresses a topic worthy of investigation, but there are some drawbacks that preclude paper acceptance.
The most important issue is the focus only on two different batches (used jars and jars used for the first time), with experiments repeated only one time. This means that all experiments were done on a single independent batch, while it is necessary to do experiments on two or three independent batches, maybe also in different times.
In addition, no details have been reported on dependent batches (technical replicates) and I can imagine that no replicates have been analyzed, as standard deviation (or any other measure of variability) is missing in figures.
This doubt on the significance of the results is a serious drawback, strongly affecting any results reported in the paper, as readers could not know if the results depend on an effective weight of the studied parameters (“age” of jars) or on a normal biological variability.
Another issue is on microbiological analyses: in dairy products, other microorganisms should be analyzed, in addition to those reported in the paper; I mean lactococci, enterococci, pseudomonads, staphylococci etc…moreover, some media used by authors are not selective as they did not contain antibiotics or other selective agents. How did they confirm the counts?
Round 2
Reviewer 6 Report
My issue on significance and replicability was not addressed, as I did not mean technical replicates but the analyses on different batches of different places and times